# Antimicrobial Resistance and Community Pharmacists’ Perspective in Thailand: A Mixed Methods Survey Using Appreciative Inquiry Theory

**DOI:** 10.3390/antibiotics11020161

**Published:** 2022-01-27

**Authors:** Rojjares Netthong, Ros Kane, Keivan Ahmadi

**Affiliations:** 1Joseph Banks Laboratories, School of Pharmacy, University of Lincoln, Beevor St., Lincoln LN6 7DL, UK; 2Faculty of Pharmaceutical Sciences, Ubon Ratchathani University, Warin Chamrab, Ubon Ratchathani 34190, Thailand; 3School of Health and Social Care, University of Lincoln, Brayford Pool, Lincoln LN6 7TS, UK; rkane@lincoln.ac.uk; 4Lincoln Medical School, University of Lincoln, Brayford Pool, Lincoln LN6 7TS, UK; kahmadi@lincoln.ac.uk

**Keywords:** antibiotic smart use programme, antibiotic stewardship, antimicrobial resistance, Appreciative Inquiry, community pharmacist, mixed method survey, Thailand

## Abstract

Global action plans to tackle antimicrobial resistance (AMR) are the subject of ongoing discussion between experts. Community pharmacists have a professional responsibility to tackle AMR. This study aimed to evaluate the knowledge of antibiotic resistance and attitudes to promoting Antibiotic Smart Use (ASU) amongst part and full-time practicing community pharmacists across Thailand. An online mixed-method survey applying Appreciative Inquiry theory was validated and conducted in 2020. Non-probability sampling was used, with online survey dissemination via social networks. A total of 387 community pharmacists located in 59 out 77 provinces seemed knowledgeable about antimicrobial resistance (mean score = 82.69%) and had acceptable attitudes towards antibiotic prescribing practices and antimicrobial stewardship (mean score = 73.12%). Less than 13% of pharmacists had postgraduate degrees. Postgraduate education, training clerkship, preceptors, and antibiotic stewardship training positively affected their attitudes. The community pharmacists proposed solutions based on the Appreciative Inquiry theory to promote ASU practices. Among these were educational programmes consisting of professional conduct, social responsibility and business administration knowledge, up-to-date legislation, and substitutional strategies to compensate business income losses.

## 1. Introduction

Antimicrobial resistance (AMR) has been recognised as an imminent threat to the global healthcare systems and the United Nation’s (UN) sustainable development goals (SDGs) [1,2,3,4]. The prevalence of AMR is on the rise, whilst new classes of antibiotics are either scarce or unavailable [5,6]. Globally, there have been action plans to tackle AMR through continuous discussions between experts, which are then fed into the international, national, and local authorities for implementation [7]. Moreover, it has been established that despite the need for a concerted effort, the challenges to combat AMR vary widely based on the geographical and economical differences that exist at the international, national and local levels [8]. The reasons for AMR are multifactorial and multidimensional [9,10,11,12]. Prescriber/dispenser behaviour, patient/client behaviour, and the governance and regulatory affairs contribute to the spread of AMR and the challenges to combat it [13,14,15,16,17]. The South East Asian region, for example, one of those worst affected by AMR, is home to low- and middle-income countries (LMICs) [18,19,20]. Prescriber and patient behaviour, availability of antibiotics and legislation of antibiotics’ sale and advertising (which is forbidden in many other parts of the world), as well as rising incomes, are the main reasons for AMR in this region [18,19]. Moreover, South East Asia has become the home to a number of upper middle-income economies such as Thailand [20,21], where not only economic development and the subsequent improvement of purchasing power, but also the legal dispensing of antibiotics without a doctor’s prescription by community pharmacists, have together exacerbated the spread of AMR [8,13,14,15,16,17,22,23,24,25]. Easy access to antibiotics and self-medication by patients led to the introduction and implementation of antibiotic smart use (ASU) by the Government of Thailand in 2007 [26]. The ASU was integrated into a national strategic plan on antimicrobial resistance (NSP-AMR 2017–2021) in 2016, and was then launched in hospital settings. The programme promoted rational antibiotic prescribing by providing financial incentives to recognise appropriate prescribing behaviour by healthcare providers in the hospital settings. The NSP-AMR 2017–2021 has had remarkable success in changing prescribers’ behaviour [14,26,27]. Following its success at the hospital level, the ASU programme was then piloted among community pharmacists with a focus on screening for pharyngitis, a common upper respiratory tract infection, in 2017 [28].

Community pharmacists have been recognised as a first port of call for healthcare advice and services because of convenient accessibility [29]. In Thailand, community pharmacists are licensed pharmacists who have completed the requirements of one of the recognised pharmacy programmes, i.e., BSc in Pharmacy, BPharm, MPharm, or PharmD, and have passed the professional licensure examination from the Pharmacy Council of Thailand (PCT) [30]. Since 2009, pharmacy education has changed to a 6-year PharmD programme replacing the 5-year BPharm programme [31]. The PharmD programme offers the graduates more specialised competencies, such as clinical skills, that are needed to deal with the day to day demands of the patients/clients [31,32]. A pharmacist could either take full employment as a community pharmacist or decide to divide their time into working for two or more employers. For example, one could work three days a week in a community pharmacy setting and two days a week in a hospital setting as a hospital pharmacist. The PCT legally authorises the community pharmacists to dispense antibiotics without a prescription [22,30,33]. Recent studies, however, have shown that more than 50% of community pharmacists did not follow the antibiotic guidelines for the management of upper respiratory infections (URIs), acute diarrhoea and simple wounds [34]. These irrational practices have directly influenced the extent of AMR in Thailand [23,24,25,34,35,36]. Therefore, the first ASU programme was set to focus on pharyngitis, being a common infectious disease that led to the community pharmacists dispensing antibiotics. It has been established that clinical practices are informed by one’s knowledge and attitudes [37,38,39,40,41]. Thus, it is crucial to evaluate community pharmacists’ knowledge of antibiotic resistance and the underlying attitudes that determine their antibiotic dispensing behaviour to further explore the main reasons behind their antibiotic prescribing practices [42].

Appreciative Inquiry (AI) theory has largely been applied to mobilise stakeholders and policy makers to encourage changes in the behaviour/practices of healthcare providers by drawing on the existing strengths and innovative ideas [43,44,45,46]. AI has been used to solve issues of complex systems that involved communication, collaboration and change in the current practices [47,48]. AI involves four interrelated phases, i.e., Discovery, Dream, Design and Destiny [46]. Discovery focuses on asking to share individuals’ best practices in their actual experiences and identifying individuals’ aspirations to develop the practices in their organisations [47,49,50]. The Dream phase magnifies ideal practices that are efficiently dominated by their own aspirations [46,51]. The Design and Destiny phases plan, implement, and sustain the desired changes and make the dream a reality and sustainable [49].

In this study, we present the quantitative and qualitative findings of our mixed method survey with a focus on exploring the factors that contribute to the current antibiotic dispensing practices amongst community pharmacists, whilst presenting the solutions proposed by the community pharmacists in regards to the four domains of the AI theory. Details of the processes involved in the design and development of the measurement tool have been presented elsewhere [46].

## 2. Results

Out of a total of 1121 part- and full-time practicing community pharmacists across Thailand who accessed the online survey tool, 387 were eligible and completed the questionnaire.

### 2.1. Socio-Demographic Characteristics of Participants

Table 1 presents the socio-demographic characteristics of the participants. The participants were from 59 out of 77 provinces in Thailand. Most were female and held a Bachelor of Pharmacy degree. Out of a total of 47 pharmacists with postgraduate degrees, 27 had a master degree in pharmacy, 3 had a master of public health, 3 had a PhD in Pharmacy, 2 had the board certified pharmacotherapy specialisation and 15 had other postgraduate degrees. Less than 95% of the participants were younger than 50, whilst slightly more than 5% were aged over 50.

### 2.2. Community Pharmacists’ Knowledge regarding Antibiotic Resistance

Table 2 presents the participants’ knowledge regarding antibiotic resistance. Overall knowledge score was 82.69% indicating a very good knowledge of antibiotic resistance. However, less than 60% of the participants chose the right answer for the following two knowledge statements: “Dispensing antibiotics without a prescription is contributing to the development of antibiotic resistance.” and “Dispensing antibiotics without a prescription is contributing to the inappropriate use of antibiotics by patients.”

Figure 1 and Figure 2 illustrate the knowledge scores and the Discovery phase scores of the participants mapped to the geographical representation of the provinces of Thailand where the participants worked. The scores are colour-coded so that green represents the highest score while red represents the lowest score. The grey colour indicates the absence of participants from that province. The participants from most of the provinces showed a high knowledge score. Mean of overall Discovery phase score, reflecting the attitue towards the current antibiotic prescribing practices, herein after referred to “attitude” scores, was 73.12%, which indicates a relatively high attitude score.

### 2.3. Community Pharmacists’ Attitude

Table 3 presents the participants’ attitude regarding antibiotic resistance; it was observed that more than 60% of participants presented good attitude in the 7 out of 12 questions. However, two questions showed high proportions of the participants answering ‘unsure’. They were “How do you rate the patients’ knowledge about antibiotic stewardship before counselling?” and “How do you rate the Thai-FDA support to implement antibiotic stewardship in community pharmacy?”.

### 2.4. Factors Associated with Attitude

Simple linear regression was used to model the correlation between Attitude as the outcome measure and a number of covariates. The selection of covariates was derived from the literature review on AMR and the challenges that have been faced by the community pharmacists when trying to tackle it.

The results of the bivariate analysis are presented in Table 4, Table 5, Table 6 and Table 7. Table 4 shows the correlation between Attitude and three of the independent variables, i.e., age, gender, and postgraduate education. The Attitude was positively affected—increased—by the age of the participants. Younger community pharmacists increased the attitude score by 0.34 SD and the effect was statistically non-significant (*p* = 0.5550). Gender, however, affected the Attitude, negatively. That is being male community pharmacists reduced the attitude score by 2.37 SD, and the effect was statistically significant (*p* = 0.0265). Postgraduate education had a relatively large positive effect on the Attitude and the effect was statistically significant (β = 5.16, *p* = 0.000664).

Table 5 shows the correlation between Attitude and antibiotic stewardship training factors. It was found that Attitude was slightly affected—decreased—by the knowledge about antimicrobial resistance. Higher knowledge decreased the attitude score by 0.01 SD and the effect was statistically non-significant (*p* = 0.823). An increase of approximately one knowledge score was associated with a decrease of 0.01 in the attitude score. However, antibiotic stewardship training experiences affected the Attitude positively. That is, being a pharmacist who had antibiotic stewardship training experience during the pharmacy course increased the attitude score by 1.78 SD, which was statistically non-significant (*p* = 0.0804). Antibiotic stewardship training experience after a graduate pharmacy degree is important, moderately increasing the attitude score by 2.07 SD, which was statistically non-significant (*p* = 0.0593). In addition, three resources to obtain antibiotic stewardship knowledge showed strong and significant association with the pharmacist attitude score. Training sessions increased the attitude score by 3.14 SD and the effect was statistically significant (*p* = 0.00485 **). Special literature statistically increased the attitude score by 2.63 SD (*p* = 0.0122). Sale representatives had a statistically positive effect on the attitude score by 2.38 SD (*p* = 0.0365). The majority of pharmacists were perceived to show antibiotic stewardship and admitted to increasing their attitude score, except the patient information leaflet showed a negative relation with pharmacists’ attitude score and the effect was statistically non-significant (*p* = 0.9140).

Table 5 shows the correlation between Attitude and antibiotic stewardship training factors. It was found that Attitude was slightly affected—decreased—by the knowledge about antimicrobial resistance. Higher knowledge decreased the attitude score by 0.01 SD and the effect was statistically non-significant (*p* = 0.823). An increase of approximately one knowledge score was associated with a decrease of 0.01 in the attitude score. However, antibiotic stewardship training experience affected the Attitude positively. That is, being a pharmacist who had antibiotic stewardship training experience during the pharmacy course increased the attitude score by 1.78 SD, which was statistically non-significant (*p* = 0.0804).

Table 6 demonstrates the correlation between Attitude and professional background factors. Attitude was slightly affected—increased—by having a BSc in Pharmacy degree. That is, being a pharmacist who had a BSc in Pharmacy degree increased the attitude score by 0.41 SD, which was statistically non-significant (*p* = 0.68). Attitude was slightly affected—decreased—by having a PharmD in pharmaceutical care. That is, being a pharmacist who had a PharmD in pharmaceutical care decreased the attitude score by 0.69 SD, which was statistically non-significant (*p* = 0.507). Attitude was slightly affected—increased—by having a PharmD in industrial Pharmacy. That is, being a pharmacist who had a PharmD in Industrial Pharmacy increased the attitude score by 0.84 SD, which was statistically non-significant (*p* = 0.651). Attitude was slightly affected—increased—by having a PharmD in pharmaceutical and health consumer protection. That is, being a pharmacist who had a PharmD in Pharmaceutical and Health Consumer Protection increased the attitude score by 0.38 SD, which was statistically non-significant (*p* = 0.957). Attitude was slightly affected—decreased—by having a PharmD in English. That is, being a pharmacist who had a PharmD in English decreased the attitude score by 0.63 SD, which was statistically non-significant (*p* = 0.929). However, clerkship affected the Attitude positively. That is, being pharmacist who had community pharmacy clerkship increased the attitude score by 1.36 SD, which was statistically non-significant (*p* = 0.3380). Being a pharmacist who had hospital pharmacy clerkship increased the attitude score by 0.06 SD, which was statistically non-significant (*p* = 0.9630). Being pharmacist who had a manufacturing clerkship increased the attitude score by 1.65 SD, which was statistically non-significant (*p* = 0.1490). Being a pharmacist who had drug registration clerkship increased the attitude score by 2.85 SD, which was statistically non-significant (*p* = 0.128). Being a pharmacist who had regulation and jurisdiction clerkship increased the attitude score by 2.81 SD, which was statistically non-significant (*p* = 0.0787). Finally, being a pharmacist who had research and development clerkship increased the attitude score by 2.22 SD, which was statistically non-significant (*p* = 0.121).

Table 7 shows the correlation between Attitude and work employment factors. Attitude was slightly affected—increased—by years of experience. Higher years of experience increased the attitude score by 0.07 SD and the effect was statistically non-significant (*p* = 0.0593). An increase of approximately one year experience was associated with an increase of 0.07 in the attitude score. Interestingly, student internship significantly affected the Attitude positively. That is, being a pharmacist who had student internship increased the attitude score by 1.21 SD, which was statistically significant (*p* = 0.000463).

### 2.5. Qualitative Results

In total, 220 of the 387 community pharmacists who responded answered open-ended questions in the survey tool.

#### 2.5.1. Regulation

A common view amongst them was that the regulations have not been consistently and fully implemented and a lack of enforcement of the regulations has implications for inconsistent practice and perceptions about their effectiveness. Pharmacists contributed their ideas for short-term steps to work towards enforcement of the regulations, such as random audits by provincial public health officers and the necessity for compulsory reports of the volume of antibiotics dispensed in community pharmacies.


*“[There are] guidelines for antibiotic use in the community pharmacy but a lack of regulation. So many community pharmacists are still improperly dispensing antibiotics.”*



*“I would like to see random official visits by provincial public health officers to evaluate the antibiotics prescription and dispensing practices of community pharmacists. These audits might help with enforcement [of the regulations].”*



*“The submission of assessment reports, detailing the amount and names of antibiotics dispensed in the community pharmacies, should be introduced. It might help to control antibiotic dispensing in the community settings.”*


#### 2.5.2. Local Guidelines

There were some suggestions about the need for local antibiotic dispensing/prescription guidelines for the treatment of infections in community pharmacies. There remains a need for effective communication for delivering the guidelines to all staff in local pharmacies nationwide.


*“I think the governmental organisation should create antibiotic dispensing/prescription guidelines for the treatment of infections in community pharmacies and then disseminate them via newsletters.”*



*“Standard practices and guidelines for rational antibiotic use in community pharmacies should be conveyed to the pharmacists throughout the country.”*


#### 2.5.3. Re-Classification

Some felt that the re-classification of antibiotics might be a key to solving inappropriate access to antibiotics and to tackling antibiotic shortage, which is increasingly becoming a problem, while others suggested that limiting antibiotic accessibility could lead to under- or ineffective treatment, contributing to increased mortality and morbidity. Re-classification was described by participants in two dimensions. First is the process to segment antibiotics into two groups differentiated by community and hospital settings. Second is a process to move antibiotics into a restricted level of classification that would require prescription. The community pharmacists reported that a significant problem was leakage of illegal antibiotics into the market especially those online.


*“Re-classification is a process of segmentation in group of antibiotics. Some can be dispensed by pharmacists in community pharmacies and some should be reserved for prescribing and dispensing only in the hospital setting for some severe infections”*



*“[There is a need for] re-classification antibiotics to a controlled drug group. This group in Thailand requires a prescription. I think it might promote rational antibiotic dispensing and use in the community settings.”*



*“Antibiotics re-classification might negatively affect patients’ experiences and outcomes including increased rates of death and disability, if they have delayed treatment of infections due to unavailable antibiotics in community pharmacists.”*



*“The first action should be to remove advertising of antibiotics from the internet, radio, or other online social media such as Lazada, Shopee, Facebook and others.”*


#### 2.5.4. Good Pharmacy Practice (GPP)

The majority of participants agreed with the statement that Good Pharmacy Practice (GPP) in the community should be utilised for regulating and standardising the quality of community pharmacists’ services including antibiotic smart use practices. Dispensing antibiotics requires community pharmacists to deliver services with counselling. There is a call for non-pharmacist staff who sell antibiotics to receive punishments. There were also suggestions that pharmacies who do not have a pharmacist on duty should have their license permission cancelled.


*“The problem is some pharmacies that lack a pharmacist on duty during working hours. Non-pharmacist can illegally sell antibiotics. It allows for the increase irrational antibiotic use in the community.”*



*“The provincial public health officers or government sectors should audit individual pharmacies or chain pharmacies where there is no pharmacist on duty. I notice that, some chain pharmacies where operate 24 h-service, without a pharmacist during the working hours.”*



*“At present, there are a lot of pharmacies that lack pharmacist on duty. At this point, the sale of antibiotics by non-pharmacist may increase the problem of antibiotic resistance.”*


#### 2.5.5. Business Pressure

In addition, there were some suggestions that community pharmacists have faced pressure about how to manage professional ethics, financial benefits and patients’ satisfaction. The difficulty of balancing the ethical aspects of professional practice and commercial pressures of community pharmacy profits while remaining professionally ethical—particularly by a group of community pharmacists who are the owners of the pharmacy—was expressed. The participants recommended interventions. Some mentioned that one strategy could be to compensate their income if they decrease the antibiotic sales by rational antibiotic dispensing. Alternative treatments, such as herbal medicines, were detailed as an alternative to antibiotic prescribing. Health promotion strategies for preventing diseases were also mentioned. A small number of community pharmacists recommended that a promotional and rewards campaign should be established to motivate rational practice in community pharmacies. This could help to reduce over-dispensing in community pharmacies. A pharmacist recommended an inventory strategy as a project that collaborated between prescribed doctors and dispended pharmacists. Community pharmacies have an inventory of antibiotics that is supported by the government.


*“I think an implementation step is a minor perform because the majority of pharmacies’ objectives are revenue and credibility. If pharmacists focus on professional ethics and have concerns about antibiotic resistance consequences*
*, they will promote rational antibiotic practices.”*



*“Antibiotic accessibility should be gradually limited in the healthcare settings including community pharmacies. Then the loss of selling antibiotics revenue may be substitute by income of other treatment choices. I suggested the promotion of the use of herbal medicine or traditional medicine for preserving income.”*



*“Promote the use of alternative products, health promotion for preventing disease.”*



*“I think an intervention should be establishing a reward system the participating pharmacy where joining the antibiotic smart use programme. An incentive might be a token as operating costs of rational antibiotic practices. Because promoting the rational use of antibiotics can lead to less of income by decreasing antibiotic sales. Thus, some pharmacies do not follow RDU because there is no benefit.”*



*“ASU is a voluntary project as well as a lack of the incentive to promote the RDU programme. There may not be a long-term intervention.”*



*“I want the system for rewarding or giving incentives to the doctor who issued a prescription with dispensing by the pharmacist at the pharmacy. And the pharmacy can operate as a government agency that dispenses ABT without fee by using ABT supported by hospitals.”*


#### 2.5.6. Public Education and Awareness

Interestingly, ideas for interventions for promoting ASU in the community have been expressed. The participants on the whole described how public education and awareness campaigns need to be tailored to specific populations. In the current context, antibiotic advertising heavily influences patients’ beliefs. Antibiotics are seen as a wonder medicine and it is in this belief that the demand for obtaining antibiotics in the community is rooted. The pharmacists noted that antibiotic advertising via radio commercial outlets along with improved regulation and consumer protection has the potential to stop inappropriate use.


*“…[There is a need to] build on the negative effects of irrational antibiotic use, and then provide guidelines to reduce the irrational antibiotic use of antibiotics across the country (for example, banned plastic bags campaign.”*



*“I think basic knowledge of antibiotic use should integrate to the primary level of education so that the general public can easily understand and have access to broad knowledge. It will help to encourage pharmacists to counsel patients to understand more easily.”*



*“I think there is a need to encourage people to gain more knowledge and understanding of the use of [appropriate] antibiotics through various media such as TV commercials, online media.”*



*“Most irrational antibiotic use is rooted in patient’s belief that antibiotics at community pharmacies and easy to obtain.”*



*“…overclaimed radio advertising about antibiotic benefits have been found to lead/cultivate audiences to misuse antibiotics.”*



*“I think that we need brochures such as antibiotic knowledge brochures for giving customers. I suggested that the brochures should easy-to-understand.”*


#### 2.5.7. Antibiotic Stewardship Training

Some participants indicated that strategies and contents of antibiotic stewardship training are a key to promoting antibiotic smart use in Thailand. They would help to achieve the promotion and standardisation of rational use of antibiotics among community pharmacists. Whilst a minority of participants mentioned that a lack of up to date information and knowledge about AMR and the ASU programme, all agreed that a concern for the community pharmacists was the lack of available materials, and resources for delivering rational antibiotic practice.


*“They may send knowledge sheets/brochures to the pharmacies or organise additional training for the pharmacies via online platforms. Because it is difficult in some areas to attend face to face training. I think we will achieve the same practice in the prescribing of antibiotics.”*



*“The provision of free accredited training throughout the country is needed.”*



*“Rational antibiotic use is promoted but it still has a lack of continuity, up to date information and lack of media support to pharmacies.”*


## 3. Discussion

To the best of our knowledge, this is the first nationwide mixed-method survey to provide contextual data on the association of part and full-time practicing community pharmacists’ knowledge of antibiotic resistance and attitudes towards current prescribing/dispensing practices by community pharmacists [34,41]. Most of the participants were from 59 out of 77 provinces in all regions of Thailand. It was generalisable and representative of community pharmacists broadly across the country.

Overall, community pharmacists’ knowledge of antibiotic resistance in this study was good. These findings align with previous research in Bangkok and Chonburi province as urban areas in Thailand using Knowledge, Attitude and Practice theory (KAP) [41]. Our study determined that less than 60% of the participants knew the consequences of dispensing antibiotics without a prescription, which can contribute to the development of antibiotic resistance and the inappropriate use of antibiotics by patients. Interestingly, only less than 4% of the participants chose the right answer to the question “Antibiotics are indicated to reduce any kind of pain and inflammation”. However, it is important to acknowledge that in the local settings in Thailand, pain and inflammation are colloquially meant to describe possible infection [50]. This question might need to be further fine tuned to improve its validity from the content point of view. These findings reflected awareness of antibiotic AMR that might be identified as a barrier to antimicrobial stewardship and AMR surveillance [34]. Antimicrobial stewardship and AMR surveillance are needed by community pharmacists for accountability and preserving antibiotic efficacy and minimising AMR [52]. Awareness campaigns in community pharmacists should urgently be implemented through education interventions [34,41].

A strength of this study was using the Discovery phase and the Dream phase of the Appreciative Inquiry (AI) theory. The Discovery phase signified the attitudes towards the actual experiences in antibiotic prescribing practices and antimicrobial stewardship in their community pharmacies. The community pharmacists expressed an acceptably high attitude. Regarding their appreciation and awareness of the current challenges of AMR in local practices, however, patients’ knowledge about antibiotic resistance and implementation to support antibiotic stewardship by the Thai-FDA were mentioned. Antibiotics are relatively inexpensive, easily obtainable and widely available due to poor implementation of demand-supply regulations, as well as some community pharmacists’ prescribing/dispensing practices, in Thailand [14]. These factors could influence the high demand and supply of antibiotics [8,13,14,15,16,17,22,23,24,25]. The community pharmacists expressed their creative ideas through the Dream phase that public education and awareness need to be tailored when implementing interventions in the Thai context. Prohibiting antibiotic advertising via radio commercials and promoting antimicrobial resistance knowledge and understanding via radio commercials and social media was suggested as a potential way to stop inappropriate demand by patients. In addition, pharmacists contributed their ideas with aspirations to improve national practices on the necessity of enforcing the regulations of antibiotic prescribing practices and supply in community pharmacies. Inspections such as random audits and compulsory antibiotic inventory reports were shared as innovative interventions through the Dream phase for modifying and standardising the community prescribing practices.

The community pharmacists mirrored the current antibiotic supply policy that most antimicrobials are widely available in pharmacies. Antibiotic re-classification was recommended for policing antibiotic supply structure. The Thai-FDA has yet to accelerate the ongoing re-classification of antibiotics. Some felt that the re-classification of antibiotics might be a key to solving inappropriate access of antibiotics and tackling antibiotic shortage. Some suggested that limiting antibiotic accessibility could lead to under or ineffective treatment, contributing to increased mortality and morbidity. Re-classification has been constructed by participants in two dimensions. The first segments antibiotics into two groups that differ by the community and hospital settings. Some antibiotics can be obtained at the community pharmacy by pharmacists’ dispensing without a prescription and others will be reserved for use only in hospital settings. The second is a process to move antibiotic groups into a restricted level, which requires a prescription. Re-classification of certain preserved antibiotic hospital and community settings and the prescribing and dispensing of antimicrobials are recommendations [27].

Preceptor training student internships, training stewardship in pharmacy curriculum, and clerkship have not previously been described. This research found a strong significant association with the pharmacists’ attitude. Interestingly, training preceptors significantly positively affected the attitude. It seems that the training preceptors had a significantly positive attitude towards rational antibiotic prescribing practices. Furthermore, they play a role in academic education through coaching future professional pharmacists. The preceptors are a part of the clinical training period as role models on the training site in the uncontrolled environment and the complexity of the task [53,54]. The community pharmacist preceptors have collaboration and involvement with the School of Pharmacy. This opportunity allows them to enhance clinical services, public health initiatives, and coordination of care within the health team. In addition, they get education material resources to help pharmacists stay up to date and design their rotation. On the other hand, the internship students provide their work allocation to allow the preceptor pharmacists to free up some duties and provide opportunities to upskill and give clinically oriented services included antibiotic stewardship in their community pharmacy.

Training stewardship in the pharmacy curriculum showed a positive association with the pharmacists’ attitudes. All clerkships contributed positive attitudes, especially regulation and jurisdiction clerkship. These community pharmacists may provide their rational practices in compliance with professional ethics. Based on legal accountability, regulation and jurisdiction clerkship may define the pharmacist’s responsibilities and provisions to ensure their practices focus on the profession rather than commercial interests [55]. According to the Pharmacy Council of Thailand (PCT), all licensed pharmacists have the authority to dispense most antibiotics without a prescription in community pharmacies [22,30,33]. Competency in community pharmacists demands that pharmacy schools develop their curricula and clerkships to support and prepare future work on community pharmacy [32]. It is crucial to intensively integrate community pharmacists’ competencies into principal pharmacy curricula or specific training and internships to ensure professional qualifications in community pharmacy practices, especially training stewardship and regulation and jurisdiction clerkships.

This research reveals that education and professional backgrounds deliver positively affected attitudes in ASU practices. Postgrad education statistically influenced the attitudes with a strong positive association. Master business administration (MBA) was the major degree that the participants had obtained; however, it was excluded from the group of postgraduate education in this study. This result seems to be consistent with other research that found that the MBA degree standalone or a dual PharmD/MBA degree are the preferred degree that the pharmacist intends to obtain in their postgrad degree [56]. Surprisingly, less than 13% of pharmacists had postgraduate degrees in directly related areas such as medicine or pharmacy; while a vignette study and the KAP study in Thailand found 10.6% and 27.83%, respectively, and the results of an international survey in 53 countries reported 26% [34,41,57]. The findings in this research determined that the majority of postgraduate degrees were Master of Pharmacy and Master of Pharmacy Programme in Community Pharmacy, which are primary degrees that community pharmacists in this research had obtained.

Postgraduate training in antibiotic stewardship has been limited. One interesting finding is that only 31.01% of community pharmacists in this research and 35% of pharmacists in a previous international survey had postgraduate training in antibiotic stewardship [57]. The previous study, using vignettes, highlighted that education and training are needed for infectious disease management, especially knowledge about antimicrobial susceptibility and use [34]. The majority of pharmacists perceived training sessions, special literature, and sale representatives as sources of antibiotic stewardship and admitted to good attitudes. In the qualitative part, some participants indicated that after acquiring a pharmacy degree, antibiotic stewardship training should regularly be implemented and up to date.

The survey results suggest that antibiotic stewardship resources strongly relate to the positive attitudes of pharmacists for rational antibiotic dispensing practices. The present results and previous studies demonstrated that the fight against AMR in community pharmacies might play within the limits of environmental work contexts to support ASU [25,58,59,60,61]. The concern for most community pharmacists was the lack of available materials, resources, surveillance systems, and infrastructure-supported collaboration with other healthcare providers for providing AMR practice [62]. Some mentioned they need a national guideline of antibiotic treatment in community pharmacies with availability and that is up to date. Effective communication in different age ranges and experiences might help achieve implementation in this setting.

Community pharmacists are healthcare providers in business settings. Based on financial benefits rather than propriety, they might face pressure to over-dispense antibiotics [63]. This can drive poor antimicrobial stewardship. In addition, customers’ demand roots in their knowledge about antibiotic use and resistance [64]. Customers’ reasons to obtain antibiotics are the belief that antibiotics can speed up recovery and many keep leftover antibiotics for future use [46]. Losing income and clients has been a concern as commercial pressures drive overprescribed antibiotics [65,66]. In this research, some community pharmacists suggested that a substitutional strategy could compensate their income if they decrease antibiotic sales by rational antibiotic dispensing. Herbal medicines were detailed as an alternative treatment. Health promotion strategies for preventing diseases were mentioned. A small number of community pharmacists recommended that a promotional and rewards campaign should be established to motivate rational practice in the community pharmacies. It could help to reduce over-dispensing in community pharmacies [67]. A pharmacist recommended an inventory strategy to collaborate between prescribed doctors and dispended pharmacists. Community pharmacies are the inventory of antibiotics supported by the government. Educational programmes consisting of professional conduct, social responsibility and business administration knowledge might balance both appropriate financial aspects and professionalism in community pharmacies. Interventions on managing professional ethics, financial benefits, and patients’ satisfaction might help to gain rational antibiotic prescribing of community pharmacists in their community setting.

Although the ASU programme has been well established in a hospital setting with remarkable success [14,26,27], the survey reveals that the ASU in community settings has not been significantly implemented yet. Less implementation, lack of regulation and enforcement capacity due to lack of up-to-date legislation can cause uncontrolled and easy access to antibiotics. Unauthorised and unregulated supply chains, including inappropriate prescription practices, contribute to self-medication and a growth of AMR [27,68]. The majority of participants agreed that Good Pharmacy Practice (GPP) in the community should be used for regulating quality of community pharmacists’ services, including ASU practice. It should cover a drug law dispensing practice that dispensing antibiotics requires community pharmacists to deliver services with counseling. Non-pharmacist staff who prescribe antibiotics is illegal and must be punished. Community pharmacies without a pharmacist on duty should have their licence revoked.

The study’s limitations are two possible aspects of bias. The online survey was generated to collect data via social media instead of a traditional post-survey to respond to the COVID-19 pandemic. In addition, the study aimed to recruit registered practicing community pharmacists from all regions of Thailand to represent community pharmacists across the whole country. A non-probability sampling technique was conducted by participants’ responses and sharing with their peers who met the eligibility criteria to be included in the study. This technique might allow participant recruitment bias associated with socio-demographic characteristics of users on online platforms, especially social media [69]. Participants appeared to be over-represented by females. It might be that women are more likely to respond to participate in both online and traditional paper surveys [34,41,70]. In addition, female pharmacists represent a higher proportion of the pharmacist population [71,72]. However, the participants were generalisable and representative of community pharmacists broadly across the country. Possible measurement bias from applying AI theory could occur due to the nature of AI definition that focuses on the strengths. Nevertheless, this research finding contributed valuable contextual data and affordable and sustainable interventions to tackle antimicrobial resistance in Thailand. In addition, this mixed-method survey allows triangulating to enhance the validity of the findings and provides a comprehensive understating of the actual practices [73].

## 4. Materials and Methods

### 4.1. Study Setting and Participants

This study aimed at recruiting the part- and full-time registered practicing community pharmacists from all regions of Thailand. Based on the 2020 census, the population of Thailand was approximately 70 million and active social media users were 52.63 million [66,68]. The survey link and recruitment advertisement in social media (FacebookTM and LineTM) were shared with almost 13,906 community pharmacies across 77 provinces [72,74].

### 4.2. Theoretical Framework and Survey Design

An extensive review of published academic and relevant literature was determined for a survey tool design [37,38,40,75,76,77,78,79]. The survey tool was developed and validated. More details can be found elsewhere [46].

Appreciative Inquiry (AI) theory was applied as a theoretical framework in creating the survey tool [46]. The closed- and open-ended questions were designed to collect information in line with the Discovery and Dream phases of the AI. Detailed information on the AI theory is published elsewhere [46]. The Discovery phase information gathering involved asking individuals to recall actual experiences of the best or the most effective practice, and also to focus on their aspirations to improve practice in their work settings [80]. The Dream phase information gathering occurred via open-ended questions, allowing expression of ideas about what the participant might have aspired to do in future practices (Figure 3).

The survey tool consisted of four sections, i.e., 24 sociodemographic questions asking gender, age, education, workplace location and experience, and resources including antibiotic learning experience; 10 knowledge statements with three response options (yes, no, don’t know) measuring knowledge of aspects of antibiotic use in Thailand including the legislation of antibiotics dispensing and awareness of antimicrobial resistance; 12 attitude questions with five-point Likert scale answer options (very good, good, unsure, poor, very poor) to assess participants’ attitudes that reflected the Discovery phase of AI theory; three open-ended questions of the Dream phase of AI to allow the participants to reflect on the current practices and to suggest their proposals for ideal practices to promote smart antibiotic use in community pharmacies in Thailand.

### 4.3. Study Design and Sampling Technique

This was a cross-sectional online survey administered on a platform called Qualtrics [81]. The survey link was disseminated nationwide via a social network of community pharmacists between February and December 2020. The survey was administered in Thai language.

A non-probability sampling technique was used, with online dissemination of the survey link to the target population via social networks. The target population was invited to participate and share the survey link, which also contained information about the target audience and the specific eligibility criteria for potential participants to ensure that social media users who were not practicing Thai community pharmacists did not participate.

### 4.4. Data Analysis

The survey tool was initially developed in the English language then translated into Thai by the lead researcher (RN) and translated back into English by a native Thai bilingual academic pharmacist. Descriptive and inferential statistics were performed using RStudio [82]. Inferential analysis used bivariate linear regression analysis. Bivariate linear regression models were performed to quantify and standardise the relationship between a predictor variable and an outcome variable. The outcome variable in this research was the attitude score. A standardised beta coefficient (β) illustrates the strength of the effect of the predictor variable to the outcome variable. The statistical significance level was *p* < 0.05.

Answers to the open-ended questions were edited before being translated from Thai into English. The editing task including minor corrections with the spellings, removing the clutter from the sentences to ensure clarity as well as improve the readability of the written answers, whilst ensuring that the main ideas and the messages were not changed or altered [83]. The edited transcriptions are said to be advantageous for translating the texts into English for analysis [83]. Qualitative data were thematically analysed using Nvivo12 software [84].

### 4.5. Ethical Considerations

The study was approved by the ethics committee at the University of Lincoln (Ethics Reference: 2019-Jul-0366), and the Institutional Review Board at Ubon Ratchathani University (Ethics Reference: UBU-REC—33/2562). Data collection was only commenced after ethical approval and securing free and informed consent of the participants.

## 5. Conclusions

This national mixed-method survey revealed the factors affecting community pharmacists’ knowledge about antibiotic resistance and attitudes about current antibiotic dispensing practices. In Thailand, the community pharmacists seem to be knowledgeable about antimicrobial resistance and have acceptable attitudes towards antibiotic prescribing practices and antimicrobial stewardship. Postgraduate education, training clerkship, preceptors, and antibiotic stewardship training positively affected attitudes in ASU practices. The solutions regarding the Appreciative Inquiry theory to promote ASU practices that were proposed by the community pharmacists ranged from educational programmes consisting of professional conduct, social responsibility and business administration knowledge, up-to-date legislation, and substitutional strategies to compensate business income.

## Figures and Tables

**Figure 1 antibiotics-11-00161-f001:**
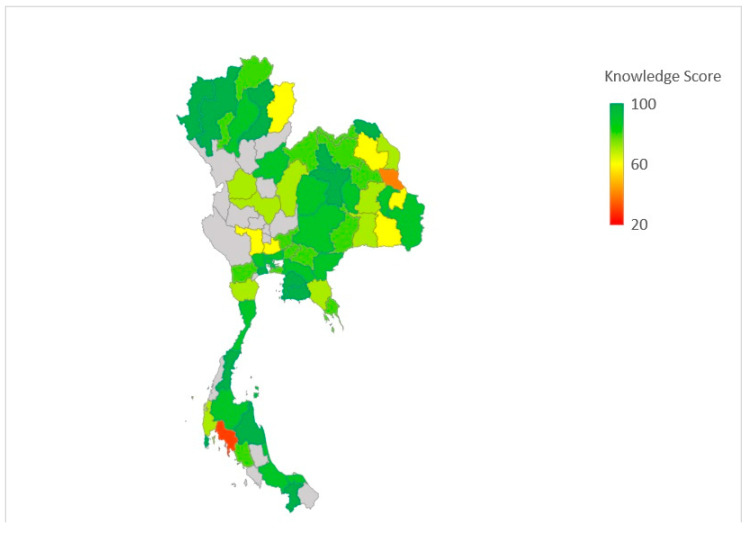
Graphical representation of participants’ Knowledge scores in the study (Powered by Bing ©GeoNames, Microsoft, TomTom).

**Figure 2 antibiotics-11-00161-f002:**
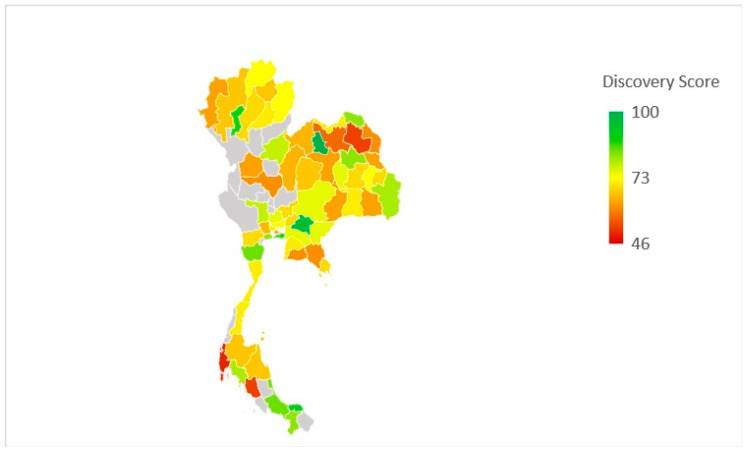
Graphical representation of participants’ Discovery phase scores in the study (Powered by Bing ©GeoNames, Microsoft, TomTom).

**Figure 3 antibiotics-11-00161-f003:**
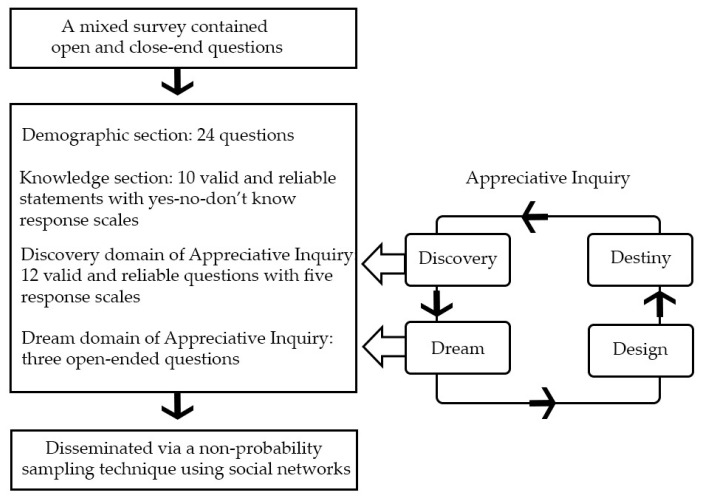
Mixed methods survey tool.

**Table 1 antibiotics-11-00161-t001:** Socio-demographic characteristics of participants.

Characteristics	N = 387 (%)
**Gender**	
Male	131 (33.85)
Female	256 (66.15)
**Age**	
Less than 30	163 (42.12)
30–39	162 (41.86)
40–49	42 (10.85)
50–59	13 (3.36)
60 and above	7 (1.81)
**Postgraduate qualification**	
Yes	47 (12.14)
No	340 (87.86)
**Location in Thailand**	
Central	149 (38.50%)
Northeastern (Isan)	117 (30.23%)
Eastern	39 (10.08%)
Southern	38 (9.82%)
Northern	22 (5.68%)
Western	5 (1.29%)

**Table 2 antibiotics-11-00161-t002:** Participants’ knowledge regarding antibiotic resistance.

Statements	Those Who Chose the Right Answer	Those Who Chose the Wrong Answer
Dispensing antibiotics without a prescription is a legal practice in Thailand.	93.80% (*n* = 363)	6.20% (*n* = 24)
Dispensing antibiotics without a prescription is a common practice among community pharmacists in Thailand.	96.38% (*n* = 373)	3.62% (*n* = 14)
Dispensing antibiotics without a prescription is contributing to the development of antibiotic resistance.	55.81% (*n* = 216)	44.19% (*n* = 171)
Dispensing antibiotics without a prescription is contributing to the inappropriate use of antibiotics by patients.	57.62% (*n* = 223)	42.38% (*n* = 164)
Antibiotics are indicated to reduce any kind of pain and inflammation.	3.36% (*n* = 13)	96.64% (*n* = 374)
Antibiotics are useful for bacterial infections.	97.93% (*n* = 379)	2.07% (*n* = 8)
Antibiotics can cause secondary infections after killing the normal flora of the human body.	93.54% (*n* = 362)	6.46% (*n* = 25)
Superbugs are microorganisms which generate antimicrobial resistance. They include bacteria, fungal, viruses or parasites.	71.32% (*n* = 276)	28.68% (*n* = 111)
Resistance DNA in bacteria can transfer to other bacteria by a virus (bacteriophage).	73.90% (*n* = 286)	26.10% (*n* = 98)
The main objective of antibiotic stewardship is the achievement of the most effective clinical outcome with the least adverse reactions.	89.92% (*n* = 348)	10.08% (*n* = 39)

**Table 3 antibiotics-11-00161-t003:** Participants’ attitude.

Questions	Very Good	Good	Unsure	Poor	Very Poor
How do you rate the implementation of local guidelines such as Antibiotic Smart Use (ASU) by the Ministry of Health, before dispensing antibiotics?	22.74% (*n* = 88)	64.60% (*n* = 250)	12.14% (*n* = 47)	0.52% (*n* = 2)	0% (*n* = 0)
How do you rate the clarity of the advice given to the patients about their dispensed antibiotics?	35.65% (*n* = 138)	61.76% (*n* = 239)	2.58% (*n* = 10)	0% (*n* = 0)	0% (*n* = 0)
How do you rate the acknowledgment of the patients’ understanding of the advice given to them about their dispensed antibiotics?	10.85% (*n* = 42)	51.68% (*n* = 200)	34.63% (*n* = 134)	2.58% (*n* = 10)	0.26% (*n* = 1)
How do you rate the answering of patients’ questions about their dispensed antibiotics?	36.18% (*n* = 140)	61.76% (*n* = 239)	2.07% (*n* = 8)	0% (*n* = 0)	0% (*n* = 0)
How do you rate patients’ satisfaction with antibiotic dispensing?	19.64% (*n* = 76)	60.98% (*n* = 236)	19.12% (*n* = 74)	0.26% (*n* = 1)	0% (*n* = 0)
How do you rate the patients’ knowledge about antibiotic stewardship before counseling?	4.65% (*n* = 18)	26.87% (*n* = 104)	32.56% (*n* = 126)	31.01% (*n* = 120)	4.91% (*n* = 19)
How do you rate the Thai- FDA support to implement antibiotic stewardship in community pharmacy?	4.39% (*n* = 17)	34.89% (*n* = 135)	45.74% (*n* = 177)	12.40% (*n* = 48)	2.58% (*n* = 10)
How do you rate engagement with antibiotic awareness campaigns?	17.83% (*n* = 69)	62.01% (*n* = 240)	17.83% (*n* = 69)	2.33% (*n* = 9)	0% (*n* = 0)
How do you rate engagement with health promotion campaigns on prevention/reduction transmission of infection?	17.05% (*n* = 66)	59.17% (*n* = 229)	20.16% (*n* = 78)	3.10% (*n* = 12)	0.52% (*n* = 2)
How do you rate collaboration (such as referral) with other healthcare professionals to implement antibiotic stewardship?	27.91% (*n* = 108)	54.01% (*n* = 209)	15.50% (*n* = 60)	2.33% (*n* = 9)	0.26% (*n* = 1)
How do you rate the relationship between clients/patients and pharmacists in regards with antibiotic stewardship?	15.76% (*n* = 61)	68.48% (*n* = 265)	13.95% (*n* = 54)	1.81% (*n* = 7)	0% (*n* = 0)

**Table 4 antibiotics-11-00161-t004:** Bivariate analysis of Attitude with socio-demographic factors such as age, gender and postgraduate education.

Variables	Attitude			
	β ^1^	SE ^2^	95% CI ^3^	*p*-Value
Age	0.34	0.57	0.78–1.46	0.5550
Male	−2.37	1.07	−4.46–−0.28	0.0265 *
Postgraduate education	5.16	1.50	2.21–8.11	0.000664 **

^1^—Parameter estimate coefficient, ^2^—[Robust] Standard Error, ^3^—Confidence interval, * *p* < 0.05, ** *p* < 0.01.

**Table 5 antibiotics-11-00161-t005:** Bivariate analysis of Attitude with antibiotic stewardship training factors.

Variables		N = 387 (%)	Attitude			
			β ^1^	SE ^2^	95% CI ^3^	*p*-Value
Knowledge		Mean = 82.96	−0.01	0.03	−0.08–0.6	0.82
Training experience during pharmacy course						
	No	211 (54.52)	1.78	1.01	−0.21–3.77	0.0804
	Yes	97 (25.07)				
	Not sure	79 (20.41)				
Training experience since degree qualified						
	Yes	120 (31.01)	2.07	1.09	−0.07–4.21	0.0593 ^#^
	No	267 (68.99)				
Sources of knowledge ^a^						
Training session		274	3.14	1.11	0.96–5.31	0.00485 **
Special literature		245	2.63	1.04	0.58–4.69	0.0122 *
Patient information leaflet		139	−0.12	1.06	−2.19–1.97	0.9140
Sales representative		106	2.38	1.13	0.14–4.60	0.0365 *
Articles in CCPE		297	0.82	1.21	−1.55–3.18	0.4980
Guidelines		258	1.51	1.07	−0.60–3.62	0.1600
Others		9	−2.74	3.37	−9.36–3.87	0.4160

^a^ Multiple answers accepted. ^1^—Parameter estimate coefficient, ^2^—[Robust] Standard Error, ^3^—Confidence interval, ^#^ *p* < 0.1, * *p* < 0.05, ** *p* < 0.01, CCPE: Center for Continuing Pharmaceutical Education.

**Table 6 antibiotics-11-00161-t006:** Bivariate analysis of Attitude with professional background factors.

Variables	N = 387 (%)	Attitude			
		β ^1^	SE ^2^	95% CI ^3^	*p*-Value
**Professional degree ^a^**					
BSc in Pharmacy	202 (52.20)	0.41	1.02	−1.58–2.41	0.68
PharmD (Pharmaceutical Care)	150 (38.76)	−0.69	1.04	−2.74–1.36	0.507
PharmD (Industrial Pharmacy)	31 (8.00)	0.84	1.87	−2.83–4.52	0.651
PharmD (Pharmaceutical and Health Consumer Protection)	2 (0.52)	0.38	7.08	−13.55–14.30	0.9570
PharmD (English programme)	2 (0.52)	−0.63	7.08	−14.55–13.30	0.9290
**Clerkship ^a^**					
Community pharmacy	329	1.36	1.42	−1.42–4.15	0.3380
Hospital pharmacy	319	0.06	1.33	−2.55–2.68	0.9630
Manufacturing	105	1.65	1.14	−0.59–3.88	0.1490
Drug registration	31	2.85	1.86	−0.81–6.50	0.1280
Regulation and jurisdiction	44	2.81	1.59	−0.31–5.93	0.0787 ^#^
Research and development	56	2.22	1.42	−0.58–5.02	0.1210

^a^ Multiple answers accepted, ^1^—Parameter estimate coefficient, ^2^—[Robust] Standard Error, ^3^—Confidence interval, ^#^ *p* < 0.1.

**Table 7 antibiotics-11-00161-t007:** Bivariate analysis of Attitude with work employment factors.

Variables	N = 387 (%)	Attitude			
		β ^1^	SE ^2^	95% CI ^3^	*p*-Value
Experience as community pharmacist					
Year	Mean = 5.59	0.07	0.08	−0.09–0.23	0.0593 ^#^
Student internships					
No	303 (78.29)	4.28	1.21	1.91–6.66	0.000463 ***
Yes	84 (21.71)				

^1^—Parameter estimate coefficient, ^2^—[Robust] Standard Error, ^3^—Confidence interval, ^#^ *p* < 0.1, *** *p* < 0.001.

## Data Availability

The data presented in this study are available on request from the corresponding author.

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
