# Peer review of "Antimicrobial Resistance and Community Pharmacists’ Perspective in Thailand: A Mixed Methods Survey Using Appreciative Inquiry Theory"

_antibiotics, 2022, doi:10.3390/antibiotics11020161_

Round 1

Reviewer 1 Report

This is an interesting manuscript on an important issue such as microbial resistance. The pharmacist, as most accessible health care professionals, are still under utilized in this public health threat. Therefore, I believe this publication would add to the body of literature on this matter. Moreover, this study is well designed and manuscript well written. Tables are appropriate and references are up to date. I only have two suggestions for the authors:

please add number of participants in the abstract

please describe the questionnaire used in the study (number of sections and items in each section)

Author Response

We are thankful to the reviewers who have kindly commented on our manuscript with the editorial number antibiotics-1568553.

After a thorough and careful consideration of the reviewers’ comments, we have presented our itemised responses in a tabular form for your kind perusal.

The changes are highlighted using the track changes option and highlighted in yellow. We have provided the line number along with page numbers for each response, where possible, to facilitate navigation through the revised manuscript.

Reviewers’ comments

Our response

Changes in revised manuscript

Reviewer 1:

This is an interesting manuscript on an important issue such as microbial resistance. The pharmacist, as most accessible health care professionals, are still under utilized in this public health threat. Therefore, I believe this publication would add to the body of literature on this matter. Moreover, this study is well designed and manuscript well written. Tables are appropriate and references are up to date. I only have two suggestions for the authors:

Reviewer 3:

The research concerns a very important problem - the knowledge and attitudes towards anitibiotic resistance among pharmacists in Thailand where pharmacists can dispense antibiotics without a prescription. This is an interesting manuscript but I believe it needs some small improvements before publishing.

Reviewer 4:

The manuscript is a great reflection of its importance and the research area of Antimicrobial Resistance. Community Pharmacists and pharmacies are the first places to receive antibiotics.

Overall the paper is well written and compiled.

We are thankful to the reviewers and appreciate the time and effort to provide feedback and insightful comments on our manuscript.

None

Abstract section

Reviewer 1:

Please add number of participants in the abstract

Reviewer 3:

Abstract - number of participants is missing, please add it

Reviewer 4:

Background in the abstract needs to rephrase and can be improved.

The abstract is not solely the reflection of the whole study, kindly rewrite it again.

We have noted that reviewers 1, 2 & 4 have made a similar comment regarding the need to rephrase the abstract.

We have added the number of participants and have rephrased the abstract.

Please refer to the abstract section.

Introduction Section

Reviewer 4:

What do mean by a part-time pharmacist?

We thank the reviewer’s comment.

We have corrected the word “a part-time pharmacist” to “part-time community pharmacist”. In addition, we have provided a description of a part-time community pharmacist in the introduction section.

Please refer to the lines 71 to 74 on page 2 under the introduction section to view the track changes.

Results Section

Reviewer 2:

In table 2, more than 96% believe that “Antibiotics are indicated to reduce any kind of pain and inflammation”. Please discuss this finding. Do pharmacists in Thailand equate antibiotics to analgesics/NSAIDS?

Reviewer 3:

2. Page 3 - numbers (percentages) in table 1 and text are different, please check it and correct it

3. Also, you have written the same data in table 1 and in text (2.1.)  Usually data is presented through the table or written in the text, so there is no reason for duplication of the results (or I just don’t see it).

Reviewer 3

4. Table 6. - When I sum numbers of pharmacists with different professional degrees, I got larger number than 387, which means that something is wrong with these numbers or some of your respondents are calculated in more than a one cathegory. Maybe it would be better to classify each participant only to the one of written groups with the education level, because in this version it looks like a mistake. The same applies to the second part of table 6. where the individual clerkships are listed.

5. Page 6, paragraph 6. - public education - it is unusual in Western contries that promotion of antibiotics is allowed, usually it is forbidden. Therefore, maybe it would be interesting to point out this difference between the legislation on the advertising of antibiotics in Western countries and Thailand, here in this part of manuscript or perhaps even better in some part of the introduction section.

Reviwer 2

In table 6, multiple responses were accepted for professional degree and clerkship. This would mean the same individuals could be analyzed twice or even thrice. Suggest to only record the highest (or most recent) qualification for each individual analyzed. The same applies to clerkship - only the most recent being included for analysis. Alternatively, such individuals should be omitted from analysis altogether.

Reviwer 3

6. Page 13, last paragraph - “less than a quarter od pharmacists had postgraduated degrees” and after that” the majority od postgrad degrees on this study were master business administration (MBA).. “I believe that MBA can’t be classiefied as postgraduate study for participants in this study, as finished postgradute study directs to conclusion that this pharmacist have additional education, and in the context of this research - additional education in the field of antimicrobial resistance. MBA is a postgraduate study in the field of economics. It can be focused on health, but it does not contribute to increasing knowledge on this topic, and pharmacies with a degree in MBA cannot be counted as those with postgraduate education, but only with graduate education. I suggest to authors either to exclude these respondents from the study or add them to the group that has the title MPharm or DPharm, depending which level of education is the highest for those pharmacists.

Reviewer 2

In their conclusion, the authors stated that “Postgraduate education... deliver positively affected attitudes....”. Logically (to someone who was not part of the study), this conclusion is only valid if MBA is not considered as a postgraduate qualification because it is neither a medical- nor a pharmacy-related qualification. So the onus is on the authors to prove that even an MBA is beneficial where combating AMR is concerned.

Reviewer 3:

Page 6, paragraph 6. - public education - it is unusual in Western contries that promotion of antibiotics is allowed, usually it is forbidden. Therefore, maybe it would be interesting to point out this difference between the legislation on the advertising of antibiotics in Western countries and Thailand, here in this part of manuscript or perhaps even better in some part of the introduction section.

We thank the reviewer for pointing this out.

We have corrected the mistake in table 2 because the numbers are a swap. We also have discussed this point in the discussion section.

We have corrected the detail in the text and rephrased the results by avoiding the duplication between table and text.

We have noted that reviewers 2 & 3 have made a similar comment regarding the professional degree and clerkship in this study. We agree that the professional degree should record the most recent. We have corrected the detail in the table 6 and the text in the paragraph. But the clerkship has not changed. As for the clerkship, a pharmacist could be through more than one clerkship; hence, we were unable to rely on the most recent clerkship.

We have noted that reviewers 2 & 3 have made a similar comment regarding the postgraduate degree in this study. We have excluded the MBA from the group of postgrad education, and then we have analysed a new number and changed the report in text and the table.

We appreciate the reviewer’s comments regarding legislation on the advertising of antibiotics in this study. We have added a sentence in a bracket “legislation of antibiotics’ sale and advertising (which is forbidden in many other parts of the world) as well as the rising income are the main reasons for AMR, in this region” .

Please refer to the table 2 page 4 under the results section and to the line to 397 to 404 page 12 under the discussion section to view the track changes.

Please refer to the lines 108 to 115 and the table 1 on page 3 under the results section to view the track changes

Please refer to the lines 201 to 216 on pages 8 and the table 6 on page 9 under the results section to view the track changes

Please refer to the lines 111 to 114, table 1 on  page 3 and lines 162 to 164 on pages 7

and the table 1 and 4 on page 8 under the results section

and lines 473 to 485 on pages 14 under the discussion section

to view the track changes

Please refer to the lines 62 to 63 on pages 2 under the introduction section to view the track changes.

Discussion section

Reviewer 3:

7. Page 14 - As antibiotics are cheap drugs, can you please describe a little bit more why are pharmacists on Thailand scared that rational antibiotics dispensing will significantly decrease their income?

Reviewer 4:

Authors must compare their results with the other reported Mixed methods studies from LMICs like the following reference may add; Khan, Faiz Ullah, Farman Ullah Khan, Khezar Hayat, Tawseef Ahmad, Amjad Khan, Jie Chang, Usman Rashid Malik, Zakir Khan, Krizzia Lambojon, and Yu Fang. “Knowledge, Attitude, and Practice on Antibiotics and Its Resistance: A Two-Phase Mixed-Methods Online Study among Pakistani Community Pharmacists to Promote Rational Antibiotic Use.” International journal of environmental research and public health 18, no. 3 (2021): 1320.

We thank the reviewer for pointing these out. We have added the discussion on the topic and followed the suggested reference.

Refer to the lines 509 to  514 on pages 14 under the discussion section to view the track changes.

Conclusion section

Reviewer 2

The last statement/sentence in the conclusion appears to be hanging. Please rephrase it.

In their conclusion, the authors stated that “Postgraduate education... deliver positively affected attitudes....”. Logically (to someone who was not part of the study), this conclusion is only valid if MBA is not considered as a postgraduate qualification because it is neither a medical- nor a pharmacy-related qualification. So the onus is on the authors to prove that even an MBA is beneficial where combating AMR is concerned.

We thank the reviewer for pointing this out. We have made the changes by rephrasing in the conclusion section.

Refer to the lines 651 to  661 on pages 18 under the conclusion section to view the track changes.

Method Section

Reviewer 1:

Please describe the questionnaire used in the study (number of sections and items in each section)

Reviewer 4:

Only 34.5% of participants have completed the survey, how about the attrition rate?

What are the reasons for the low response rate?

How about the active internet users in Thailand? If the number is available please put it in the methods section.

The methods section needs more explanation.

We really appreciate the very positive comments regarding the design of the study.

We have expanded on this point to explain number of sections and items in each section about the questionnaire and cite the previous study to see the questionnaire. We have added the number of Thai population and active social medias’ users.

Please refer to the lines 560 to 596 and lines 579 to 588 on pages 16 under the method section to view the track changes

Reviewer 2 Report

The language is generally acceptable, but further refinements by an English language expert is advised.

In table 2, more than 96% believe that "Antibiotics are indicated to reduce any kind of pain and inflammation". Please discuss this finding. Do pharmacists in Thailand equate antibiotics to analgesics/NSAIDS?

In table 6, multiple responses were accepted for professional degree and clerkship. This would mean the same individuals could be analyzed twice or even thrice. Suggest to only record the highest (or most recent) qualification for each individual analyzed. The same applies to clerkship - only the most recent being included for analysis. Alternatively, such individuals should be omitted from analysis altogether. 

In their conclusion, the authors stated that "Postgraduate education... deliver positively affected attitudes....". Logically (to someone who was not part of the study), this conclusion is only valid if MBA is not considered as a postgraduate qualification because it is neither a medical- nor a pharmacy-related qualification. So the onus is on the authors to prove that even an MBA is beneficial where combating AMR is concerned.

Also, the last statement/sentence in the conclusion appears to be hanging. Please rephrase it.

Author Response

(The authors gave the same response as above.)

Reviewer 3 Report

Thank you for the opportunity to review this paper. The research concerns a very important problem - the knowledge and attitudes towards anitibiotic resistance among pharmacists in Thailand where pharmacists can dispense antibiotics without a prescription.  This is an interesting manuscript but I believe it needs some small improvements before publishing:

1. Abstract - number of participants is missing, please add it

2. Page 3 - numbers (percentages) in table 1 and text are different, please check it and correct it 

3. Also, you have written the same data in table 1 and in text (2.1.)  Usually data is presented through the table or written in the text, so there is no reason for duplication of the results (or I just don't see it).

4. Table 6. - When I sum numbers of pharmacists with different professional degrees, I got larger number than 387, which means that something is wrong with these numbers or some of your respondents are calculated in more than a one cathegory. Maybe it would be better to classify each participant only to the one of written groups with the education level, because in this version it looks like a mistake. The same applies to the second part of table 6. where the individual clerkships are listed.

5. Page 6, paragraph 6. - public education - it is unusual in Western contries that promotion of antibiotics is allowed, usually it is forbidden. Therefore, maybe it would be interesting to point out this difference between the legislation on the advertising of antibiotics in Western countries and Thailand, here in this part of manuscript or perhaps even better in some part of the introduction section.

6. Page 13, last paragraph - "less than a quarter od pharmacists had postgraduated degrees" and after that " the majority od postgrad degrees on this study were master business administration (MBA).. " I believe that MBA can't be classiefied as postgraduate study for participants in this study, as finished postgradute study directs to conclusion that this pharmacist have additional education, and in the context of this research - additional education in the field of antimicrobial resistance. MBA is a postgraduate study in the field of economics. It can be focused on health, but it does not contribute to increasing knowledge on this topic, and pharmacies with a degree in MBA cannot be counted as those with postgraduate education, but only with graduate education. I suggest to authors either to exclude these respondents from the study or add them to the group that has the title MPharm or DPharm, depending which level of education is the highest for those pharmacists.

7. Page 14 - As antibiotics are cheap drugs, can you please describe a little bit more why are pharmacists on Thailand scared that rational antibiotics dispensing will significantly decrease their income?

Author Response

(The authors gave the same response as above.)

Reviewer 4 Report

I am thankful to the Antibiotics-MDPI for inviting me to review the given manuscript. The manuscript is a great reflection of its importance and the research area of Antimicrobial Resistance.  Community Pharmacists and pharmacies are the first places to receive antibiotics. Following are my comments for authors

  1. Background in the abstract needs to rephrase and can be improved.
  2. The abstract is not solely the reflection of the whole study, kindly rewrite it again.
  3.  What do mean by a part-time pharmacist?
  4.  Only 34.5% of participants have completed the survey, how about the attrition rate? 
  5. What are the reasons for the low response rate?
  6. How about the active internet users in Thailand? If the number is available please put it in the methods section.
  7. The methods section needs more explanation.
  8. Authors must compare their results with the other reported Mixed methods studies from LMICs like the following reference may add; Khan, Faiz Ullah, Farman Ullah Khan, Khezar Hayat, Tawseef Ahmad, Amjad Khan, Jie Chang, Usman Rashid Malik, Zakir Khan, Krizzia Lambojon, and Yu Fang. "Knowledge, Attitude, and Practice on Antibiotics and Its Resistance: A Two-Phase Mixed-Methods Online Study among Pakistani Community Pharmacists to Promote Rational Antibiotic Use." International journal of environmental research and public health 18, no. 3 (2021): 1320.
  9. Overall the paper is well written and compiled. 

Author Response

(The authors gave the same response as above.)
